# PML Body Biogenesis: A Delicate Balance of Interactions

**DOI:** 10.3390/ijms242316702

**Published:** 2023-11-24

**Authors:** Sergey A. Silonov, Eugene Y. Smirnov, Irina M. Kuznetsova, Konstantin K. Turoverov, Alexander V. Fonin

**Affiliations:** Laboratory of Structural Dynamics, Stability and Folding of Proteins, Institute of Cytology, Russian Academy of Sciences, St. Petersburg 194064, Russia; e.smirnov@incras.ru (E.Y.S.); imk@incras.ru (I.M.K.); kkt@incras.ru (K.K.T.)

**Keywords:** PML-bodies, SUMO/SIM, intrinsically disordered protein, TRIM domain, liquid–liquid phase separation, membraneless organelle, biomolecular condensates

## Abstract

PML bodies are subnuclear protein complexes that play a crucial role in various physiological and pathological cellular processes. One of the general structural proteins of PML bodies is a member of the tripartite motif (TRIM) family—promyelocytic leukemia protein (PML). It is known that PML interacts with over a hundred partners, and the protein itself is represented by several major isoforms, differing in their variable and disordered C-terminal end due to alternative splicing. Despite nearly 30 years of research, the mechanisms underlying PML body formation and the role of PML proteins in this process remain largely unclear. In this review, we examine the literature and highlight recent progress in this field, with a particular focus on understanding the role of individual domains of the PML protein, its post-translational modifications, and polyvalent nonspecific interactions in the formation of PML bodies. Additionally, based on the available literature, we propose a new hypothetical model of PML body formation.

## 1. Introduction

The discovery of promyelocytic leukemia protein (PML) is associated with the research on acute promyelocytic leukemia (APL) in the 1990s, where it was shown that the disease is accompanied by a chromosomal translocation t(15;17) in approximately 90% of cases [1,2,3]. The resulting oncogenic fusion protein consists of two proteins—retinoic acid receptor alpha (RARα) and PML, located on chromosomes 17 and 15, respectively. Since the previously unknown region of chromosome 15 was designated as myl (from myelocytic leukemia), one of the outdated synonyms for PML is MYL [1].

Thirty years of intensive research on PML in numerous laboratories worldwide have revealed that PML is a multifunctional protein that plays a key role in both physiological and pathological cellular processes. These processes include tumor suppression or oncogene function depending on tumor context, DNA damage response, cell migration, epigenetic composition of chromatin, apoptosis, senescence, neoangiogenesis, antiviral defense, and hematopoietic stem cell maintenance [4,5,6,7,8]. Several comprehensive review articles have focused on the roles of PML isoforms in protein–protein interactions [2,9,10,11,12,13], as well as in infectious diseases [14,15,16].

The coding PML gene spans approximately 53 kbp and consists of nine exons (Figure 1). Due to alternative splicing of the C-terminus end, the PML protein is represented by seven major isoforms: five with nuclear localization (PML-II to PML-VI), one cytoplasmic (PML-VII), and one nuclear/cytoplasmic isoform (PML-I). The first four exons (418 amino acids (aa), RBCC motif) are common to all isoforms and encode structurally conservative domains: Proline-rich (1–45 aa), RING (45–105 aa), B1-box (124–166 aa), B2-box (184–229 aa), and coiled-coil (229–323 aa) [17]. The presence of the RBCC motif classifies PML as a member of the TRIM (tripartite motif-containing) protein family, with two synonymous names—TRIM19 and RNF71 (RING finger protein 71). In addition to the seven major isoforms, approximately five minor isoforms have been identified, which differ from the main PML isoforms I–VI by exon loss. Based on this, the following PML subtype subgroups have been proposed: subtype A—loss of exon 5; subtype B—loss of exons 5 and 6; subtype C—loss of exons 4, 5, and 6 [6,9]. It is worth noting that PML multifunctionality is due to the presence of different functional isoforms. As an example, on the one hand, PML is overexpressed in triple-negative breast cancer (TNBC), where it exhibits oncogenic functions, promoting tumor metabolism, maintenance of cancer stem cells, and metastasis [8], while on the other hand, PML-IV suppresses the proliferation and self-renewal of TNBC cells [18]. PML, identified in the fused protein of retinoic acid receptor (RAR)/PML in acute promyelocytic leukemia, is one of the early cancer targets for targeted drug therapy. While decreased expression of the PML protein is associated with tumor progression in many types of cancer, some tumors exhibit unexpectedly high levels of PML [19]. Thus, the conflicting properties of PML may be indicative of specific tumor cell characteristics and the poorly understood functions of individual PML isoforms.

The largest isoform, PML-I (882 aa), contains a purposeful nuclear export signal (NES, 704–713 aa, exon 9) at its C-terminal region, which is consistent with observations of this isoform’s localization in both the nucleus and the cytoplasm (within early endosomes) [6,13,20,21].

PML isoforms are incorporated into PML bodies in the cellular nucleus. The nuclear localization signal (NLS) is responsible for the import of PML into the nucleus. Mutations in the NLS have been shown to result in the formation of bodies for all PML isoforms. For PML-III, PML-IV, PML-V, and PML-VII isoforms with mutated NLS, the formation of toroid-shaped bodies co-localized with the late endosome marker LAMP1 has been observed [21]. This effect suggests a universal mechanism of body formation that is independent of nuclear localization and initiates during protein expression in the endoplasmic reticulum (ER).

At the present time, it is known that the nuclear localization sequence (NLS) in exon 6 is absent in two isoforms: PML-VII (PML-VIIb according to nomenclature [6]) and PML-VIb. While exclusive cytoplasmic localization has been demonstrated for the PML-VIIb isoform [21], practically nothing is known about PML-VIb. It is worth noting that the full-length PML-VI can partially localize to the nuclear lamina [22], and the amino acid sequence of the PML-VI isoform differs from other isoforms by only eight amino acid residues (GRE**RNALW**). Considering that the C-terminal region of PML-VIb contains a similar sequence, **RNALW** (419–423 aa), it can be assumed that this protein has a similar localization. The functional features of the PML-VIb isoform also remain unknown. However, they are of high interest, especially considering that this isoform is represented by two mRNA molecules, differing in the length of the untranslated region (Table 1).

## 2. PML Regulation at the mRNA Level

Nature efficiently organizes its cellular processes. Cells are capable of transporting, localizing, and concentrating specific compounds at their site of action, making them more accessible. This allows them to organize and regulate key life reactions, enhance their efficiency, and control incoming and outgoing fluxes [23].

According to the modern view, the spatial and temporal regulation in living cells is based on the liquid–liquid phase separation (LLPS) of biopolymers [24]. This process segregates proteins and nucleic acids into specialized micron-sized bodies known as biomolecular condensates or membraneless organelles (MLOs), which lack membranes but exhibit liquid-like properties [25]. MLOs are capable of rapid assembly and disassembly in response to stimuli and play an important role in cellular spatial–temporal organization [26]. The change in biopolymers’ molecular concentration is the main driving force of LLPS [27]. One of the checkpoints for protein concentration control is the mRNA level, which has numerous regulatory regions throughout its sequence [28]. One such region in the PML protein is the 5′ untranslated region (5′-UTR), which is approximately 140 nucleotides in length [9], is shared among all isoforms, and contains a functional internal ribosome entry site (IRES) that allows cap-independent initiation of protein translation [29]. Although PML mRNA is expressed in most cell types and the PML gene is rarely mutated or deleted from the genome, PML protein expression is often suppressed in human cancer and frequently correlates with tumor grade and progression [30,31].

It is hypothesized that translation mediated by IRES represents a mechanism of selective upregulation of apoptosis-related protein expression when cells are under stress conditions, such as oxidative stress and genotoxic stress, which are known to suppress global protein translation [32]. For the PML protein, it has been shown that IRES activity can be induced in a TNFα-dependent manner through p38 MAPK and MNK1, leading to the accumulation of PML protein and the formation of PML bodies. It is worth noting the proposition that the p38-MNK1-PML network regulates TNFα-induced apoptosis in breast cancer cells and TNFα-mediated inhibition of migration and capillary tube formation in endothelial cells [33]. Additionally, it should be noted that IRES is present in all PML isoforms, indicating that TNFα (or other unknown IRES inducers) induces the translation of all isoforms in one way.

It was demonstrated that the 5′-UTR of PML mediates oncogenic K-RAS-induced translation of PML mRNA through mTOR/eIF4E. However, the precise mechanism of this process, particularly the specific region within the 5′-UTR, has not been determined [34].

Alternative splicing of primary PML transcripts results in multiple PML isoforms, each of which possesses **unique** 3′-UTRs (Figure 1). The longest 3′-UTRs are found in the PML-I (2805 nt), PML-V (1738 nt), and PML-VI (1258 nt) isoforms. It has been shown that miR-1246 (micro RNA) can target the mRNA of PML-I and decrease the abundance of this isoform in cells [35]. Interestingly, this has been demonstrated in colorectal cancer cell lines that utilize microvesicles containing miR-1246 and TGF-β for effective modulation of the tumor microenvironment, including enhanced angiogenic activity of endothelial cells, which can uptake these microvesicles [35].

It is interesting to note that besides impacting cell cycle progression and proliferation, miR-1246 may also influence the stemness and resistance of cancer cells to therapeutic agents [36]. In addition to miR-1246, it has been discovered that miR-24 and miR-133 also target the 3′-UTR of PML-I mRNA and decrease the expression level of PML-I protein in primary human endothelial cells under normal culture conditions [30]. Unexpectedly, miR-24, but not miR-133, has the ability to activate the translation of PML-I mRNA in cells cultured under conditions of cellular starvation. It has also been shown that miR-24 inhibits angiogenesis in vitro and ex vivo through a PML-dependent pathway [30]. In a recent study on circular RNAs (circRNAs) in chronic lymphocytic leukemia, hsa-miR-337–3p was found to interact with the 3′UTR of PML-I and lead to a decrease in the concentration of this isoform in cells [37].

The findings obtained to date regarding the 3′-UTR of PML-I suggest that the expression levels of all PML isoforms may be regulated at the mRNA level through their unique 3′-UTR sequences. This process is theoretically expected to involve a complex, multifactorial control mechanism depending on the internal state of the cell.

## 3. Roles of SUMO-SIM Interaction

It is known that PML is a scaffold protein in nuclear PML bodies (PML NB). PML NBs are present in most mammalian cell lines and tissues [38]. Depending on the cell type, cell cycle phases, and differentiation, one cell may contain approximately 5–30 PML NBs ranging in size from 0.1 to 2 μm [39,40]. One of the early models of PML NB formation proposed that PML NB assembly depends on sumoylation at three lysine sites (K-65, K-160, and K-490), leading to the maturation of PML bodies through subsequent SUMO-SIM interactions with their partners [41,42]. It has been shown that the sumoylation of K-65 occurs only after the sumoylation of K-160, with both K-65 and K-160 being sumoylated by PIAS proteins [43]. PIAS1-mediated sumoylation of PML promotes its interaction with CSNK2A1 (CK2) and phosphorylation at Ser-565 (within the VVVISSSEDSDEANSSSR region), which in turn initiates ubiquitin-mediated degradation of PML. Although phosphorylation at Ser-565 has been identified as the main site, there may be additional phosphorylation sites crucial for this process [44]. It is worth noting that exon 7, encoding Ser-565, is absent in the PML-VII isoform, indicating that the degradation pathway of PML-VII does not involve CK2. Determining which of the possible degradation pathways [45] is the main one for PML-VII remains to be elucidated.

The three-site sumoylation model was complemented by studies [46,47] in which PML protein with three mutations at K-65, K-160, and K-490 was capable of forming PML NBs. In the study [46], a construct with the three mutations K-65, K-160, and K-490, along with an additional mutation in the SIM site disrupting SUMO-SIM interactions at the SIM site of PML, was also investigated. Transfection of cells with this construct did not abolish the formation of PML bodies but led to a disruption in their shape. Subsequently, apart from the three main sumoylation sites of PML, other mono- and poly-sumoylation sites (K-209, K-226, K-337, K-380, K-394, K-400, K-401, K-476, K-478, K-487, K-497, and K-616) have been proposed [48]. However, their role in PML body formation remains poorly understood.

Recent biochemical and biophysical in vitro studies have shown SUMO-SIM-dependent liquid–liquid phase separation (LLPS) and provided evidence that phase separation in PML bodies can be regulated by SUMO polymers recruiting SIM-containing proteins [49]. Furthermore, it has been discovered that SUMO-SIM-dependent LLPS can contribute to the formation of APBs (ALT-associated PML bodies) [50,51]. These findings were revolutionary, as they demonstrated for the first time the role of weak interactions and the potential for protein concentration in one place through LLPS.

It should also be noted that many SUMOylated proteins are ubiquitinated in PML-nuclear bodies by RNF4, RNF111, or SUMO-targeted ubiquitin ligases (STUbL), resulting in their degradation by the 26S proteasome [52].

Taking the above into consideration, it can be concluded that although individual SUMO-SIM interactions are relatively weak, they can have a profound impact on the structure of PML bodies at different levels: altering the activity, localization, and stability of both PML and PML body proteins, initiating the formation of macromolecular assemblies, acting as a kind of intermolecular “glue”, and inducing phase separation, potentially leading to both assembly and disassembly of the PML bodies [53,54].

One well-known function of PML-nuclear bodies is their role as hubs for post-translational modifications and quality control centers for nuclear proteins [53]. PML-nuclear bodies involve enzymes that participate in multiple post-translational modifications, primarily the SUMO-conjugating enzyme E2 UBC9, a key enzyme conjugating SUMO to E2, which is thought to promote partner protein SUMOylation [55].

In 1999, it was shown that SUMO-1 covalently modifies PML both in vivo and in vitro and that the modification is mediated by direct or indirect interaction between UBC9 and PML via the PML RING domain [56]. SUMOylation of UBC9 at K-49 contributes to its localization in PML nuclear bodies [57]. It is proposed that E3 SUMO ligases typically use the RING domain to interact with UBC9. However, it was shown that the RING domain of PML does not directly bind to UBC9 in vitro. Instead, the PML B1-box domain exhibits UBC9-binding activity [58]. It is worth noting that the putative UBC9-binding site on PML coincides with the UBC9-binding site for its upstream E1 enzyme, which transfers SUMO to UBC9. The overlap of these two binding sites suggests that UBC9 cannot simultaneously interact with its E1 and E3 partners [58]. However, this contradiction can be resolved by the presence of two adjacent B1-box domains in the PML structure, possibly in the form of dimers, which is consistent with the hypothetical model proposed in the Section 5 (“Hypothetical icosahedral PML structure”).

Another proposed model was associated with cooperative action between the RBCC motifs of PML, which were hypothesized to prepare the platform for SUMOylation and SUMO-SIM interactions, thus promoting polymerization and phase separation [49]. However, clear cooperation between these processes has not been shown, and SUMO-SIM interactions are now more commonly considered facilitators for recruitment. Since a notable amount of research on PML has been linked to PML-RARα and its response to arsenic trioxide (ATO), which is considered a last resort treatment for acute disease, a PML body formation model reflecting studies in this area has emerged. This model suggests that (i) oxidative stress induces cross-linking of PML monomers through the formation of disulfide bonds; (ii) weak non-covalent interactions of RBCC promote oligomerization of non-SUMOylated PML proteins; (iii) UBC9-mediated (poly-)SUMOylation of PML occurs, leading to SUMO-SIM interactions; (iv) the process culminates in stabilization of the outer shell, possibly involving liquid–liquid phase separation mechanisms [5,49]. This model also presented a contradiction, as several studies demonstrated that the C-terminus end of PML can form nuclear PML-nuclear bodies in the absence of the RBCC motif and endogenous PML expression in cells [42,59,60]. Apparently, weak protein interactions in PML bodies are also important, alongside strong interactions, for the biogenesis/formation of PML bodies in cellular space.

## 4. PML Domains and Their Role in PML Body Formation

TRIM family proteins are united by the presence of conserved RING, B-box, and coiled-coil (CC) domains. In addition to their involvement in numerous cellular processes, these proteins also participate in innate antiviral response pathways [61]. Despite the discovery of dimerization and higher-order assemblies, the atomic details of the tertiary and quaternary structures of these proteins remain incompletely understood. The following provides the reader with key information on the crystal structures of individual PML domains, the structural aspects of different mutant variants of this protein in cells, and the structural features of related TRIM family proteins that share amino acid sequence similarities with PML.

### 4.1. Analysis of PML Sequences Using AlphaFold

We analyzed the sequences of the major PML isoforms using the AlphaFold2 algorithm [62] (Figure 2). In accordance with expectations, all PML isoforms align perfectly with each other in the RBCC domain region (34–418 aa). For PML-II, a predicted alpha-helix in the region Ala(650)-Leu(672) is shown, corresponding to residues 652–681, which is responsible for the localization of PML-II to the nuclear lamina [21]. An interesting result of prediction is the discovery of an alpha-helix in the region 421–430 aa for PML-II, III, IV, V, and VI, which corresponds to the unexplored exon 5 of the PML gene. Despite the low per-residue model confidence score (pLDDT) for the predicted alpha-helices, these structural features can be confirmed or disproven in future studies.

### 4.2. RING Domain PML: Dimerization and Tetramerization

The crystallographic structure of PML RING domains at high resolution revealed the ability of PML RING domains to assemble into a tetrameric torus-shaped complex [55]. The tetramer can be divided into two types of constituent dimers (Figure 3B): (i) L73-RING dimers, where interactions occur among amino acids (P71, L73, H74, C88, C91, P94, A93), with significant hydrophobic interactions between L73 residues; (ii) F52-RING dimers, where interactions take place between the FQF loop (F52, Q53, F54) and a hydrophobic pocket (L70, L81, W95), with phenylalanine F54 of the F52-RING dimer sandwiched between the side chains of K65 (within the same subunit) and K68 (of the second subunit in the dimer). Additionally, an intermolecular disulfide bridge (C66-C66) was discovered in the crystal structure of the tetramer [55].

According to the analysis of one of the earliest obtained monomeric PML RING domain structures (1BOR.pdb), it was suggested that the FQF loop interacts with the L73 pocket of its own subunit [55]. On the other hand, since in the tetrameric RING structure, the FQF loop interacts with the hydrophobic pocket of another subunit, a hypothesis emerged regarding the possible activation (priming) of PML monomer prior to dimerization [4]. However, the obtained structural data for the PML RING monomer (2MWX.pdb) revealed a discrepancy, as there was no detected interaction between the FQF loop and its corresponding L73 pocket (Figure 4A) [64]. Analyzing the crystallization conditions, we noted that both the 1BOR and 2MWX structures were obtained under the same pH and buffer conditions, but the 1BOR sample consisted of a synthetic peptide refolded in the presence of Zn^2+^, while 2MWX was expressed and purified directly from *E. coli* in the presence of 20 μM Zn^2+^ [64]. We decided to examine the positions of F52 and L73 in the generated AlphaFold2 prediction model and discovered that the FQF loop could be concealed within a hydrophobic pocket between the B1- and B2-box domains (F52 is in proximity to L174 and L178, Figure 4B). Thus, we propose that L73 dimerization may represent the main dimerization for PML body formation, while the FQF loop’s involvement in oligomerization could serve as a hydrophobic remnant to maintain subsequent macromolecular structure.

An important step towards understanding the biogenesis of PML bodies was taken in an experiment where co-transfection of wild-type PML plasmids was performed with mutant variants of PML RING domains (HA-F52/54E and HA-L73E) [55]. It was found that the presence of the HA-L73E mutant completely blocked the assembly of PML bodies, while the presence of HA-F52/54E did not hinder the formation of PML bodies but reduced the number of PML bodies per cell. This observation underscores the primary importance of the correct assembly of L73-RING dimers, which appear to be the initial step in PML body formation. Since PML knockout MEF cells still formed PML bodies, albeit in smaller quantities, with the mutant PML HA-F52/54E, it can be assumed that the formation of F52-RING dimers, as well as the subsequent formation of RING tetramers in the assembly of PML bodies, should be called into question.

### 4.3. B1- and B2-Box Domains PML: Dimerization and Oligomerization

The structure of the B1- and B2-box domains of the TRIM family can facilitate dimerization and potential oligomerization [65]. The crystallographic structure of the PML B1-box domain (6IMQ.pdb) revealed the presence of dimers formed through interactions involving W157 and F158 residues [40]. In the W157 dimer, contact is facilitated by two sets of hydrophobic interactions between W157 and F152/F138, while in the F158 dimer, contact is mediated through hydrophobic side chain interactions located at the center of the dimer interface. It is worth noting that according to the crystal structure data (Figure 3B), the dimers (6IMQ.pdb) are antiparallel, with the N- and C-termini of the protein oriented in opposite orientations. In addition to the antiparallel structure of the B1-box domain, a parallel B1-box crystal structure has also been reported (2MVW.pdb) [66]. Furthermore, the parallel structure of B-box domains is predominantly observed in the crystal structures of the TRIM family [65].

Similar to the experiments on RING domains, PML mutants W157E, F158E, and I122P/V123P were studied [40]. Exogenous PML mutants were subjected to arsenic trioxide (ATO) treatment in PML knockout HeLa cell lines, which resulted in different effects on PML NB formation. Unlike the L73E mutant (RING domain), which exhibited irreversible impairment of PML NB assembly, the damage to PML NB assembly observed in the case of the F158E or W157E/F158E mutants (B1-box domain) was reversed after ATO treatment [40]. The lack of restoration effect on PML NB upon ATO treatment in the L73E mutant supports the evidence of a cooperative oligomerization mechanism between the RBCC domains [4]. It is hypothesized that tetrameric structures may potentially form between the dimers of the W157 and F158 B1-box domains, ultimately leading to oligomerization and the establishment of an extensive B1-box network [40]. Currently, this proposition is still open to debate.

Unfortunately, there is no information about the PML B2-box domain crystal structure. However, experimental data regarding the properties of this domain have been obtained. Sulforaphane (SFN) is renowned for its anti-tumor activity and is widely used as an antioxidant. Nonetheless, the understanding of the mechanisms underlying SFN’s action on cells remains poorly explored. SFN is present in cruciferous vegetables such as broccoli, watercress, Brussels sprouts, and cabbage [67]. LC–MS/MS analysis revealed that SFN modifies several cysteine residues (C57, C60, C129, C189, and C204) in PML, leading to the inhibition of PML NB formation and nuclear localization blockade [68]. To assess the impact of each residue, mutants with corresponding cysteine substitutions were generated. In the case of SFN treatment, mutant PML C57A and C60A displayed redistribution of small (0.03–0.5 μm) and medium-sized (0.5–2.1 μm) PML body populations without significant change in the total number of bodies per cell. The only cysteine residue critical for PML body formation proved to be C204, located in the B2-box domain (RBCC) [68].

This suggests that processes occurring prior to C204 (i.e., RING di- and tetramerization, B1-dimerization/oligomerization) are necessary but insufficient for the assembly of native PML bodies. Based on the structure, it can be inferred that C204 of the B2 domain may contribute as a key element in the formation of B2-box dimers and B1-box/B2-box oligomerization of PML. Supporting the theory of potential parallel dimerization of B2-box, related proteins from the TRIM family have presented similar structural profiles: MuRF1/TRIM63 (3DDT.pdb) [69]. TRIM5α (5K3Q) [70], TRIM54 (3Q1D), and TRIM28 (2YVR). Structural comparisons of known B2-box structures of TRIM proteins using UCSF Chimera demonstrated significant structural conservation among them (Figure 5).

The comparison conducted showed that the B1- and B2-box domains of PML have very similar spatial structures, indicating their potential to possess similar properties [58]. Considering the similarity between the B1 and B2-box domains, it can be hypothesized that both the B1-box and B2-box domains are capable of dimerization and interaction with partner proteins. It can be assumed that if the E1 protein interacts with the RING domain, E2 (UBC9) interacts with the B1-box domain, and E3 interacts with the B2-box domain, then the PML protein may have a universal “adapter” function. This hypothesis can be either proven or disproven in the future.

### 4.4. Proline-Rich and Coiled-Coil Domain

The structure and role of the proline-rich domain of PML remain poorly understood at present. There is evidence of phosphorylation in this region by HIPK2 and ERK1/2 MENDELEY CITATION PLACEHOLDER 0. It is speculated that this phosphorylation promotes sumoylation and is necessary for the pro-apoptotic activity of PML after DNA damage. Due to the predicted disorderliness of this domain, it can be assumed that it is essential for protein–protein interactions and signal transduction [22]. Considering that during antiparallel dimerization of PML RING domains, proline-rich tails may hinder the formation of subsequent B-box domain structures, we propose that the proline-rich domain may act as a potential protector against antiparallel dimerization of PML RING domains and serve as a potential “controller” for proper parallel dimerization of RING domains and subsequent assembly of PML nuclear bodies.

It is known that the proteins of the TRIM family are linked by the presence of conserved RING, B-box, and coiled-coil (CC) domains. In addition to their involvement in various cellular processes, these proteins participate in innate antiviral pathways [61]. Despite the discovery of higher-order dimerization and assemblies, the atomic details of the tertiary and quaternary structures of these proteins remain significantly unexplored. Recent crystallographic and biochemical analyses of the CC domain of several TRIM family proteins (TRIM5α, TRIM20, TRIM25, and TRIM28) have revealed that TRIM proteins dimerize in an antiparallel fashion through the CC domains, positioning the N-terminal RING domains at opposite ends of the dimer, while the C-terminus, which typically interacts with partner proteins, may be located at the center [65]. The dimers include central hendecad repeats and a symmetric pattern of flanking heptads, which appear to be conserved throughout the TRIM family [61,73,74,75].

Experimental evidence has revealed the spatial arrangement of coiled-coil dimers of TRIM5α in hexagonal arrays, allowing for the recognition of the capsid lattice of HIV-1 and the restriction of retroviral replication [74]. It is worth mentioning that PML, as a component of PML nuclear bodies, also plays a role in the life cycle of HIV-1: when transcriptionally silent, HIV-1 co-localizes with PML nuclear bodies, while during transcriptional activation, the provirus is released from PML-rich regions into the nucleoplasm [76]. It was previously believed that PML proteins strongly restrict the early stages of HIV-1 and other lentivirus infections when expressed in mouse embryonic fibroblasts (MEFs). However, recent studies have shown that PML knockout did not affect the permissiveness of these cells to HIV-1 infection, indicating that, unlike TRIM5α, PML is not a restrictive factor for HIV-1 in human cell lines [77].

In a study of the primate cytomegalovirus protein IE1, which consists of a globular core (IE1_CORE_) surrounded by internally disordered regions, it was demonstrated that IE1_CORE_ binds to PML through its coiled-coil (CC) domain and can also interact with TRIM5α [78]. The authors suggest that IE1_CORE_ sequesters PML and possibly other members of the TRIM family through structural mimicry, utilizing an extended binding interface formed by the CC domain.

Under normal conditions, TRIM5α localizes to the cytoplasm; however, treatment with leptomycin B (LMB) was able to retain this protein in the nucleus and observe its co-localization with PML nuclear bodies [79].

The aforementioned experiments involving PML and TRIM5α suggest not only similar functionality and conformation of their coiled-coil domains but also close structural features, such as the hexagonal arrays of PML protein oligomers. These findings allow us to speculate in the “Hypothetical Icosahedral PML Structure” section.

Recent studies on coiled-coil (CC) domains have shown that CC-containing proteins, such as centrosomal proteins pericentrin [80], spd-5 [81], and centrosomin [82], could potentially undergo liquid–liquid phase separation (LLPS). In silico modeling has demonstrated that the physical characteristics of CC domains are sufficient to govern protein LLPS [83]. It has been revealed that CC domains forming trimers and tetramers have a significantly higher propensity for LLPS compared to CC domains forming dimers [83]. This suggests that the state of multimerization has a greater impact on LLPS than the number of CC domains within a protein.

Little is known about the structure of the CC domain of PML (Figure 2); however, it is worth noting an interesting fact that the CC domain of PML represents a minimal structural determinant required for the transformation of RAR into an oncogenic protein (PML-RARα fusion) [84].

Considering the presence of RING, B1-box, and B2-box oligomerization in PML, we hypothesize that during the maturation of PML nuclear bodies, there may be the formation of structures in which the coiled-coil (CC) domains contribute to oligomerization and liquid–liquid phase separation (LLPS).

### 4.5. Disordered C-Terminal Domains

PML isoforms share a common N-terminus and a unique C-terminus. It is known that the C-terminal domain of each PML isoform critically participates in the formation of PML bodies [51]. The prevailing model of PML body formation in the last decade proposed three maturation phases: (i) PML multimerization, driven by the RBCC N-terminal domain, leading to the formation of solid protein aggregates of various sizes; (ii) reorganization of PML and assembly into ordered hollow spheres; and (iii) recruitment of protein partners into PML-NBs. A point of controversy in the formation mechanism was the requirement of SUMO-SIM interactions for the transition from phase I to phase II [85]. Studies with PML-IV have demonstrated the necessity of SUMO-SIM interactions [46], while the opposite has been shown for PML-III [85].

Among the six major nuclear PML isoforms in human cells, only PML-I and PML-V are conserved between humans and mice [20]. Despite being expressed in small amounts, PML-V can make a significant contribution to the structure, integrity, and functionality of PML bodies. Low expression levels of PML-V may even be necessary for normal nuclear body function, as expression above a certain threshold can lead to insoluble aggregates [42]. It has been postulated that PML-V acts as a scaffold in PML bodies, as it is particularly stable during fluorescence recovery after photobleaching (FRAP) analysis [86]. PML-V has been shown to be a good substrate for RNF4, suggesting that PML-V is heavily modified by SUMOylation compared to other PML isoforms [87]. Thus, the increased residence time observed for PML-V may be associated with its high levels of SUMO modification. For PML-V, it has been established that its C-terminus contributes to the assembly, maintenance, and structural stability of nuclear PML bodies. The motif responsible for these unique properties is a hypothetical α-helical sequence (591–611 aa), which is only present in PML-V [42]. Mutations destabilizing the α-helical motif, such as Arg-599/603 to Pro, almost completely abolished the ability of the C-terminus of PML-V to form nuclear bodies [42].

It has been shown that PML-I, II, and VI accumulate in the cytoplasm after arsenic treatment, whereas PML-III, IV, and V do not accumulate [87]. Additionally, arsenic-induced degradation of PML isoforms was dependent on the ubiquitin E3 ligase RNF4. Therefore, the variable C-terminal domain influences the rate and location of PML isoform degradation.

It is supposed that PML-IV is the only PML isoform that can activate p53 and induce cellular senescence upon overexpression, making it one of the extensively studied isoforms [59]. It has been shown that the unique region of PML-IV exon 8 (referred to as 8b in the literature) can interact with ARF and promote global SUMOylation, particularly of p53 [88]. This 8b region also plays a critical role in the formation of PML bodies, as its deletion disrupted the toroid-like structure despite the presence of SIM, whereas the addition of 8b to the C-terminus of PML-VI induced the formation of these structures [88]. An interesting finding is that 8b has the ability to interact with SUMO, and it can bind to both SUMO-1 and SUMO-2, with its interaction with SUMO-1 being even stronger than the SIM region of PML. Thus, to some extent, 8b may function as an additional SIM motif, which could explain the unique functional characteristics of PML-IV [59,88].

In 2012, the first evidence emerged that the C-terminal domains of PML-II and PML-V are capable of regulating the structural and functional properties of PML bodies [42]. It was demonstrated that the C-terminus of PML-V interacts with the C-terminus of PML-III and PML-IV to form bodies in endogenous PML knockout cells. According to yeast two-hybrid analysis [42], the C-terminus of PML-II does not interact with itself, yet it still forms PML bodies in PML knockout MEF cells. This discrepancy can be resolved if the C-terminus of PML-II is capable of liquid–liquid phase separation (LLPS). Recent findings of the PML-II and PML-V C-terminal domains revealed their ability to form dynamic droplet-like compartments independently of the ordered RBCC PML N-terminal motif [22,60]. Sequence analysis of PML isoforms showed that certain isoforms, primarily PML-II, have a tendency for LLPS due to their polyampholytic properties and C-terminus disordered structure. It is speculated that such structures can serve as “nucleators” for functionally active PML bodies, providing the necessary concentration of PML isoforms for intermolecular interactions between PML monomers [60]. Considering the ability of the C-terminal region of PML-V to interact with the C-terminus of other isoforms, it can be speculated that the PML-V C-terminus acts as a unique “carrier/concentrator” of other isoforms within PML bodies through LLPS.

Various types of stress stimuli induce interactions between PML and the nucleolus, resulting in the formation of PML multi-protein nucleolar structures (PML-NDS) (nucleolar caps) [89]. Only two isoforms of PML (PML-I and PML-IV, and under genotoxic stress—PML-V) effectively localize to the nucleolus, with PML-I being essential for the nucleolus localization of endogenous PML isoforms [89]. Interestingly, specifically in PML-I, a unique ordered C-terminal region has been predicted among other PML isoforms (Figure 2) [89]. Dynamic structural changes in the interaction between PML and the nucleolus are closely associated with the inactivation and reactivation of RNA polymerase I (RNAP-I) transcription, which is presumably a fundamental process in maintaining the precise balance between DNA repair mechanisms, greatly impacting genome integrity and the aging process [90]. It is worth noting that PML proteins do not spontaneously accumulate in the nucleolus, indicating the presence of internal mechanisms that restrain this process. It is known that PML-IV overexpression in human diploid fibroblasts and MEF induces cellular senescence [91]. However, PML-IV does not induce cellular senescence in PML knockout MEF cells with oncogenic Ras expression, suggesting the requirement of other isoforms for this process [89]. PML-V is suitable for this role since, as mentioned earlier, its C-terminus can interact with PML-IV. Taken together, the observed features of PML-I and PML-IV demonstrate that the targeting of PML to the nucleolus during aging may not be a side effect but rather a precise regulation of this process.

The PML-VII isoform forms toroidal structures in the cytoplasm of cells despite the absence of a SIM motif in its composition and a large, disordered C-terminus [21]. It has been shown that PML-VII co-localizes with LAMP1, indicating the association of PML-VII with late endosomes. Upon detailed examination of the acquired images in the study [21], a hypothesis emerges suggesting the coverage of PML-VII around late endosomes, which makes the investigation of the functions and roles of PML-VII in the cytoplasm highly interesting.

In summary of this section, the available data to date provide compelling evidence that the C-terminus of PML isoforms is crucial in regulating the localization, assembly dynamics, and functions of PML bodies.

## 5. Hypothetical Icosahedral PML Structure

We propose a hypothetical model (Figure 6) based on the following premises: (i) The PML RING domain is capable of dimerization (possible tetramerization); (ii) The B1-box and B2-box domains can undergo dimerization (potential oligomerization); (iii) SUMO-SIM may facilitate liquid–liquid phase separation (LLPS) with subsequent “gluing” of PML proteins; (iv) The coiled-coil domains of PML may participate in LLPS and, akin to other TRIM family members, potentially form antiparallel dimers; (v) The C-terminus of PML-II and PML-V are potentially capable of LLPS, with the C-terminus of PML-V able to interact with the C-terminus of other PML isoforms; and (vi) Crystal structures of RBCC domains from other TRIM protein family members, particularly the hexameric form of TRIM5α, are known. It is worth noting that a “Model of the symmetric tetramer” was proposed for TRIM5α, which was consistent with the obtained data in the presence of a tetramerization region [73], a role that may be played by the RING and B1- and B2-box domains in the case of PML.

In the framework of the hypothetical model, we speculate that the initial step in PML nuclear body formation involves the condensation of PML monomers through LLPS effects and weak interactions. The second step involves monomer dimerization through the RING domain. The third step involves simultaneous dimerization through the coiled-coil domains and the B1-box and B2-box domains. The resulting structure adopts a truncated, icosahedral-like arrangement. In this structure, we propose the presence of six central elements of PML, in contrast to TRIM5α [74], which has three central elements. Assuming three central elements for PML would result in one RING domain always being available for dimerization, whereas this is not the case with six central elements. Furthermore, the presence of four coiled-coil domains in the connecting element may correlate with previous in silico data, suggesting the contribution of oligomerization of these domains to LLPS [83]. It is worth noting that within this model, an individual dimer may represent a PML isoform interacting with PML-V, whose C-terminus has a propensity for LLPS and interactions with the C-terminal regions of other PML isoforms.

Such a structure has several advantages: (i) it solves the issue of rapid exchange of the internal composition of mature PML bodies and nucleoplasm by forming regulated pores in the framework structure. This would be difficult if there existed a dense layer of concentrated oligomeric protein; (ii) it allows for a large flow and quantity of proteins to exchange with the nucleoplasm (provided they meet the conditions for passage through the pores), noting that at least 120 proteins can physically interact only with PML isoforms and the PML interactome continues to expand [2]. It is also worth mentioning the development of a novel technology called SUMO-ID, which successfully identified 59 high-confidence SUMO-dependent interactors of the PML proximal proteome [92]; (iii) throughout the assembly process of PML bodies, they can undergo post-translational modifications and interact with their partners both in the shell and inside the PML bodies; (iv) individual links of the hexameric structure remain open for interactions, which can be important, including for proteasomal degradation of PML proteins [31] and for the reduction/enlargement of the hexamer structure.

In 2017, Li Chuang and colleagues [59], using super-resolution microscopy (SIM), demonstrated potential PML hexameric structures (Figure 6). However, it remains unclear whether the obtained images are artifacts of sample imaging or one of the first documented evidence of the existence of hexameric structures. Furthermore, to date, similar images have emerged using super-resolution microscopy in other studies [93,94]. According to the data obtained from correlative electron and super-resolution microscopy, the thickness of the PML shell is approximately 50–100 nm [38,87]. The distance between coiled coils (218–346 aa), based on a hypothetical model constructed using the Alphafold2 (AF-P29590-F1), is approximately 175 Å (~17.5 nm). According to the model, cavities at 90 nm and 150 nm may occur upon the dissociation of individual hexamers from the hexameric shell (Figure 6).

## 6. Conclusions

It is known that phase transitions and liquid–liquid phase separation (LLPS) play a crucial role in the structuring of intracellular space. Recently, new findings on the ability of C-terminal domains of PML-II and PML-V isoforms to form dynamic liquid droplet compartments in PML knockout HeLa cells have demonstrated the importance of these weak interactions in the formation of PML bodies. While it remains unknown whether the C-termini are involved at every stage of PML body formation, their role in the condensation of PML monomers at the initial stage of body assembly becomes evident. The obtained data on the crystal structures of the RING and B-box domains of PML, as well as the RBCC domains of the TRIM protein family, have provided insights into the assembly mechanism of native PML bodies. Despite the hypothetical model proposed in this review, it remains unclear how RBCC domains interact with each other and what 3D structures they actually form. In the near future, studies on the structures of potential coiled-coil domain dimers and B2-box domains of PML, as well as the detailed examination using super-resolution microscopy, are expected, which will further facilitate the prediction of PML body structure. Additionally, the study of the impact of different PML isoforms, post-translational modifications, and protein–protein interactions on the processes of PML NB assembly is an extensive field that warrants exploration. Furthermore, LLPS involving SUMO-SIM, coiled-coil, or C-terminal domains of PML-II and PML-V isoforms further complicates the understanding of these processes. As stated in one of the articles [5], dedicated to PML, ‘catch me if you can,’ with each passing year, the scientific community is getting closer and closer to catching and understanding these processes.

## Figures and Tables

**Figure 1 ijms-24-16702-f001:**
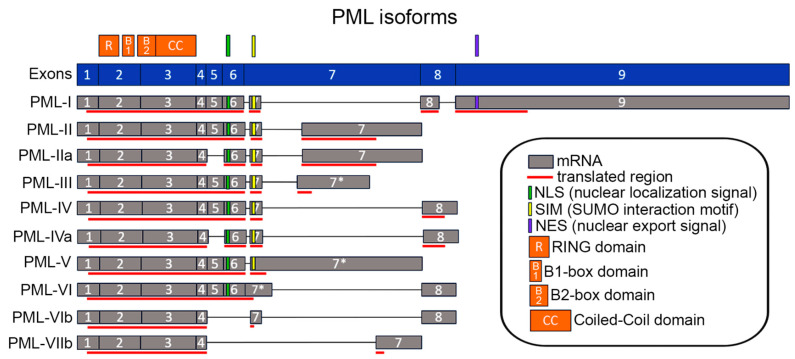
PML isoforms and mRNA structures are produced by alternative splicing. The PML gene exons are shown in blue, with rectangle sizes proportional to the nucleotide sequence lengths. The alternative splicing-derived mRNAs are represented in gray. The mRNA-translated regions are shown in red, while the untranslated regions (UTRs) are indicated in the remaining regions. The asterisks indicate that intron retention occurs in PML-III, -V, and -VI [6]. The alignment data were obtained using the NCBI database.

**Figure 2 ijms-24-16702-f002:**
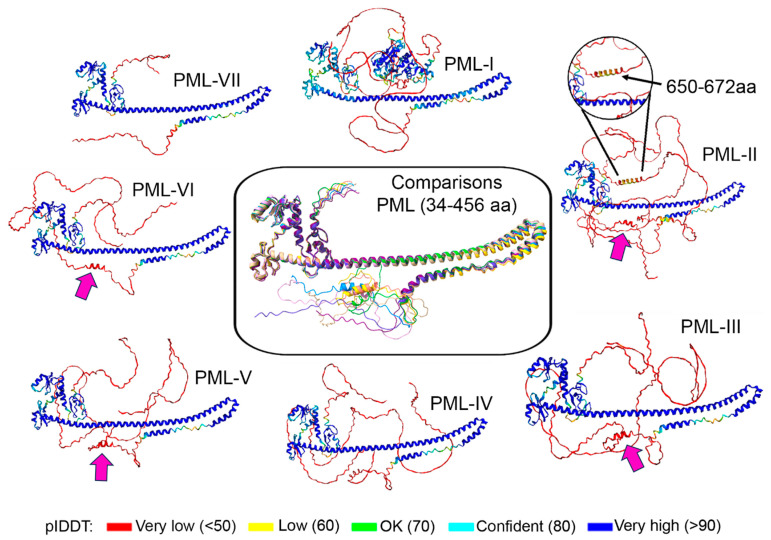
AlphaFold2 prediction analysis for the major PML isoforms. The center panel shows the overlay of structures ranging from amino acid residues 34–456, generated in UCSF ChimeraX [63]. For PML-II (top right panel), an identified alpha-helix in the region of 650–672 aa is shown. Pink arrows indicate the predicted alpha-helix at amino acid residues 421–430, corresponding to exon 5 of the PML gene.

**Figure 3 ijms-24-16702-f003:**
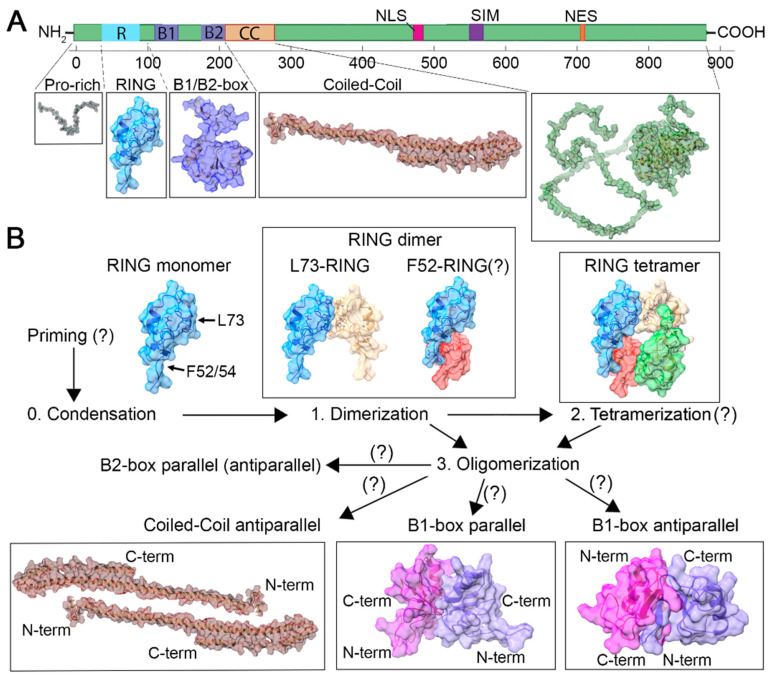
Known structural models based on crystallographic studies and AlphaFold2 prediction (AF-P29590-F1)—Panel (**A**). Proposed pathways of PML protein oligomerization—Panel (**B**). Structures depicted using UCSF ChimeraX [63]: RING monomers, dimers, and tetramers (5YUF.pdb); B1-box antiparallel dimer (6IMQ.pdb); B1-box parallel dimer (2MVW.pdb). The arrows indicate potential pathways of PML oligomerization.

**Figure 4 ijms-24-16702-f004:**
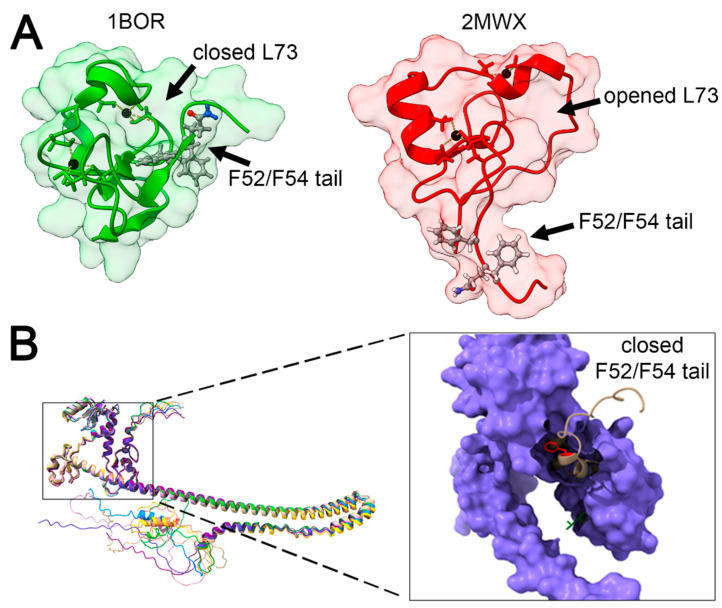
Structure of the PML RING domain. Panel (**A**): Comparison of crystal structures obtained from the Protein Data Bank (1BOR.pdb—green, 2MWX.pdb—red). Panel (**B**): Position of the FQF loop (F52/F54 tail, shown in red) in the predicted AlphaFold2 structure of PML. It is demonstrated that the FQF loop is concealed within the hydrophobic pocket of the B1- and B2-box domains. Images were generated using UCSF ChimeraX [63].

**Figure 5 ijms-24-16702-f005:**
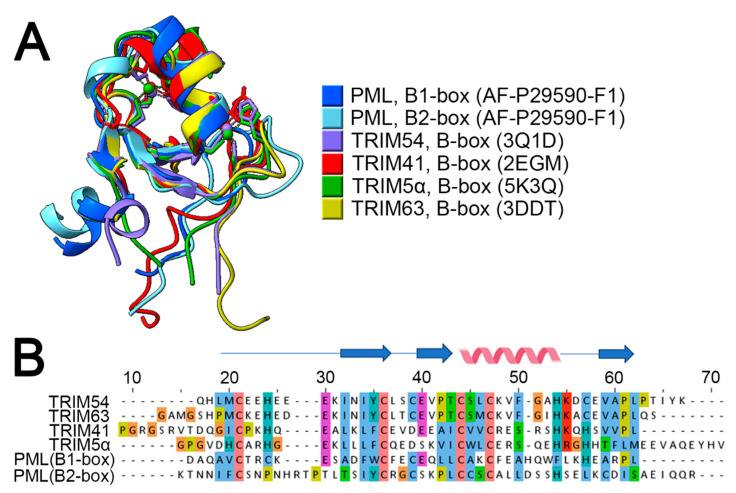
Comparison of TRIM family B-box domains. Panel (**A**)—Overlay of PML B1- and B2-box domain structures (blue, cyan—AlphaFold2 prediction) and crystal structures of B-box domains: TRIM63 (3DDT.pdb), TRIM5α (5K3Q.pdb), TRIM41 (2EGM.pdb), and TRIM54 (3Q1D.pdb). Panel (**B**)—Sequence alignment of B-box domains, with β-sheets and α-helices displayed on top. Colors correspond to the default Clustal color scheme. Structure comparison was performed using ChimeraX, sequence alignment was performed using ClustalW [71], and visualization was conducted using Jalview [72].

**Figure 6 ijms-24-16702-f006:**
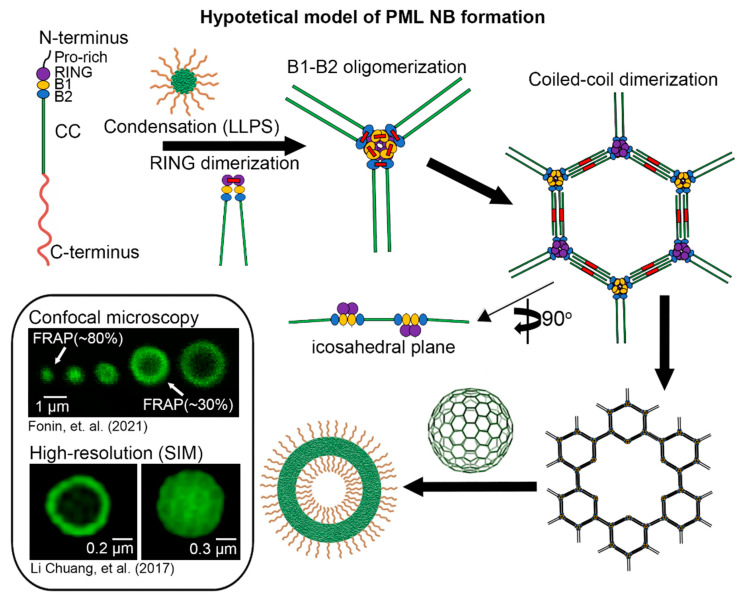
Hypothetical model of PML body formation. Shown are the stages of PML protein maturation through liquid–liquid phase separation (LLPS), followed by RING, B1–B2 box domain (B1, B2) dimerization, and coiled-coil (CC) domain dimerization to form the hexameric structure. Confocal microscopy data from our group’s studies [22,60] are presented. It is shown that smaller PML bodies exchange their components with the nucleoplasm more rapidly (as evidenced by fluorescence recovery after photobleaching (FRAP) recovery rate of ~80%), while larger, ‘mature’ PML bodies exchange more slowly (FRAP recovery of about ~30%). Super-resolution SIM microscopy data are taken from an open research article by Li Chuang and colleagues [59].

**Table 1 ijms-24-16702-t001:** Nomenclature of PML protein isoforms.

#	Ensembl Transcript Name	Jensen et al. Name [17]	TRIM Name	NCBI Name	UniProt Name	Length, aa	Mw, kDa	UniProt Match	CCDS
1	PML-201	PML-I	TRIM19 alpha	Isoform 1	PML-11	882	97.6	P29590–1	CCDS10255
2	PML-202	PML-II	TRIM19 kappa	Isoform 9	PML-2	829	90.7	P29590–8	CCDS10257
3	PML-203	PML-IIa	-	Isoform 11	PML-13	781	85.7	P29590–13	CCDS10258
4	PML-204 PML-217	PML-VIb	TRIM19 iota TRIM19 eta	Isoform 7	PML-14	423	47.6	P29590–14	CCDS45300
5	PML-205	PML-VIIb	TRIM19 theta	Isoform 8	PML-7	435	48.6	P29590–10	CCDS10256
6	PML-206	PML-IV PML-X	TRIM19 zeta	Isoform 6	PML-4	633	70.0	P29590–5	CCDS45297
7	PML-207	PML-V	TRIM19 beta	Isoform 2	PML-5	611	67.5	P29590–2	CCDS45298
8	PML-208 PML-221	PML-VI	TRIM19 epsilon	Isoform 5	PML-6	560	62.0	P29590–4	CCDS45299
9	PML-211	PML-IVa	TRIM19 lambda	Isoform 10	PML-12	585	65.0	P29590–12	CCDS58386
10	PML-215	PML-Ia	-	-	PML-11	834	92.6	P29590–11	-
11	PML-220	PML-III	-	-	PML-31	641	70.4	P29590–9	-
12	-	PML-IIg	TRIM19 gamma	-	PML-8	824	90.2	P29590–3	-
13	PML-210	-	-	-	-	568	62.9	-	-

## Data Availability

Not applicable.

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
