# Peer review of "PML Body Biogenesis: A Delicate Balance of Interactions"

_ijms, 2023, doi:10.3390/ijms242316702_

Round 1

Reviewer 1 Report

Comments and Suggestions for Authors

This review provides a comprehensive overview of the different PML isoforms, their structural domains, and how this protein regulates protein-protein interactions to facilitate PML body self-assembly and interaction with other proteins. The authors also present an interesting model for the PML body, which is partly based on recent super-resolution microscopy data. The review is well-written, well-structured and the cited literature is appropriate.

Author Response

We are grateful to Reviewer 1 for carefully reading our work and providing valuable comments

Reviewer 2 Report

Comments and Suggestions for Authors

This review by Silonov et. al covers significant aspects of PML body biogenesis. I believe is well composed and the message is clear. I simply have some specific comments that may improve this manuscript:

Paragraft 32-38 and lines 94-95: In the last 10 years, PML has been shown to be overexpressed in triple negative breast cancer where it has a role in stemness maintenance and metastasis promotion. This function has been proven glioma as well, a role that opposes its typical function as a tumor suppressor protein. This concept should be discussed in the review.

Line85. The section heading "results" has to be rethought because, given this is a review publication, the authors primarily collect and analyze publicly available data rather than presenting results. Consequently, the section entitled “2.9. Hypothetical icosahedral PML structure” can be a separate section in the manuscript.

Line 87. Liquid–liquid phase separation (LLPS) is a novel principle for explaining the precise spatial and temporal regulation in living cells. LLPS compartmentalizes proteins and nucleic acids into micron-scale, liquid-like, membrane less bodies with specific functions, which were recently termed biomolecular condensates. The concept of LLPS should be properly introduced.

Line 538. A subset of PML proximal proteome that is SUMO-dependent has been identified using SUMO-ID technique.  

A very recent paper showed that PML modulates epigenetic composition of chromatin, I think that this is a topic that is important to discuss.

Author Response

We are grateful to Reviewer 2 for carefully reading our work and providing valuable comments.

Paragraf 32-38 and lines 94-95: In the last 10 years, PML has been shown to be overexpressed in triple negative breast cancer where it has a role in stemness maintenance and metastasis promotion. This function has been proven glioma as well, a role that opposes its typical function as a tumor suppressor protein. This concept should be discussed in the review.

Reply

We added the discussion of this topic to the text

Line85. The section heading "results" has to be rethought because, given this is a review publication, the authors primarily collect and analyze publicly available data rather than presenting results. Consequently, the section entitled “2.9. Hypothetical icosahedral PML structure” can be a separate section in the manuscript

Reply

We changed the sequence of chapters and subsections.

Line 87. Liquid–liquid phase separation (LLPS) is a novel principle for explaining the precise spatial and temporal regulation in living cells. LLPS compartmentalizes proteins and nucleic acids into micron-scale, liquid-like, membrane less bodies with specific functions, which were recently termed biomolecular condensates. The concept of LLPS should be properly introduced.

Reply

We added LLPS concept to the text

Line 538. A subset of PML proximal proteome that is SUMO-dependent has been identified using SUMO-ID technique. 

Reply

We identified SUMO-dependent interactors and added this information into text

A very recent paper showed that PML modulates epigenetic composition of chromatin, I think that this is a topic that is important to discuss.

Reply

We added discuss by this topic to the text